# Prevention of Postprandial Hyperglycemia by Ophthalmic Nanoparticles Based on Protamine Zinc Insulin in the Rabbit

**DOI:** 10.3390/pharmaceutics13030375

**Published:** 2021-03-12

**Authors:** Saori Deguchi, Fumihiko Ogata, Takumi Isaka, Hiroko Otake, Yosuke Nakazawa, Naohito Kawasaki, Noriaki Nagai

**Affiliations:** 1Faculty of Pharmacy, Kindai University, 3-4-1 Kowakae, Higashi-Osaka, Osaka 577-8502, Japan; 2045110002h@kindai.ac.jp (S.D.); ogata@phar.kindai.ac.jp (F.O.); 1611610016p@kindai.ac.jp (T.I.); hotake@phar.kindai.ac.jp (H.O.); kawasaki@phar.kindai.ac.jp (N.K.); 2Faculty of Pharmacy, Keio University, 1-5-30 Shibakoen, Minato-ku, Tokyo 105-8512, Japan; nakazawa-ys@pha.keio.ac.jp

**Keywords:** insulin, nanoparticle, postprandial hyperglycemia, instillation, polyacrylic acid

## Abstract

Postprandial hyperglycemia, a so-called blood glucose spike, is associated with enhanced risks of diabetes mellitus (DM) and its complications. In this study, we attempted to design nanoparticles (NPs) of protamine zinc insulin (PZI) by the bead mill method, and prepare ophthalmic formulations based on the PZI-NPs with (nPZI/P) or without polyacrylic acid (nPZI). In addition, we investigated whether the instillation of the newly developed nPZI and nPZI/P can prevent postprandial hyperglycemia in a rabbit model involving the oral glucose tolerance test (OGTT). The particle size of PZI was decreased by the bead mill to a range for both nPZI and nPZI/P of 80–550 nm with no observable aggregation for 6 d. Neither nPZI nor nPZI/P caused any noticeable corneal toxicity. The plasma INS levels in rabbits instilled with nPZI were significantly higher than in rabbits instilled with INS suspensions (commercially available formulations, CA-INS), and the plasma INS levels were further enhanced with the amount of polyacrylic acid in the nPZI/P. In addition, the rapid rise in plasma glucose levels in OGTT-treated rabbits was prevented by a single instillation of nPZI/P, which was significantly more effective at attenuating postprandial hyperglycemia (blood glucose spike) in comparison with nPZI. In conclusion, we designed nPZI/P, and show that a single instillation before OGTT attenuates the rapid enhancement of plasma glucose levels. These findings suggest a better management strategy for the postprandial blood glucose spike, which is an important target of DM therapy.

## 1. Introduction

Plasma glucose (PG) levels are maintained within a narrow range throughout the day, with a normal fasting PG level in the range of proximately 60–110 mg/dL. In general, peak PG levels occur after a meal, and the rapid elevation in PG levels is known as the “blood glucose spike”. Postprandial PG levels may reflect an early stage in type 2 diabetes mellitus (DM), known as “postprandial diabetes” [1]. Moreover, postprandial hyperglycemia enhances the risk of diabetes complications in type 2 DM [2,3,4,5,6,7]. Epidemiologic studies suggest significant associations between the enhanced risk of cardiovascular events and elevated PG levels [8,9] and elevated PG levels 2 h after a meal are associated with an enhanced risk of mortality [2]. Thus, postprandial hyperglycemia plays a decisive role in the onset of chronic metabolic disorders in both pre-DM and DM patients [10,11]. Therefore, more treatment strategies for early prevention and intervention of DM are particularly important.

Insulin (INS) replacement is a selective therapy for type 2 DM patients and used as an effective therapy to regulate PG levels in type 1 DM patients [12]. INS treatment is also one potential strategy for alleviating postprandial hyperglycemia. Approximately 20–30 years ago, studies were conducted in dogs [13], cats [14], and rabbits [15] on the systemic delivery of INS using INS eye drops. However, the results of those studies showed that the delivery of INS solution through the conjunctival sac and nasal cavity could not achieve sufficient plasma INS levels for the regulation of PG [16] due to low INS bioavailability (BA) [13,17] and its short duration on the cornea [18]. For these reasons, INS preparations are still administered as injections. On the other hand, it is possible that a more recent novel ocular drug delivery system (DDS), such as one involving nanoparticles, might improve the problems of low BA and short duration and might be useful for the prevention of postprandial hyperglycemia.

Nanoparticles represent a comfortable DDS with very prolonged action in the ophthalmic field since smaller particles are better tolerated by patients than larger particles [19]. It has been reported that the effect of reducing the particle size in ocular DDS can result in a three-fold increase in the post-instillation concentration of a drug [19]. We have designed solid nanoparticles (NPs) containing indomethacin, tranilast, and minoxidil by using the bead mill method, which is one of the top-down approaches [20,21,22], and used them to prepare ophthalmic dispersions [21,22]. Moreover, we have found that particles approximately 150 nm in size result in increased ophthalmic absorption of a drug (indomethacin) via the multiple energy-dependent endocytosis system in the ocular surface [23]. Additionally, the combination of an in situ gelling system and drug NPs can achieve high ocular drug BA by enhancing the contact time of the NPs [23,24,25]. In addition, Yamamoto et al. [26] reported that the combination of an INS solution, sodium glycocholate (NaGC), and polyacrylic acid (PAA) can maintain the INS level on the conjunctival sac after instillation and produce a significant increase in the contact time of the INS solution at the absorption site (conjunctival sac and nasal cavity), although they were unable to achieve practical utility. In this study, we attempted to design nanoparticles of protamine zinc insulin (PZI) by the bead mill method and prepare ophthalmic dispersion containing PZI-NPs (nPZI). In addition, we created ophthalmic formulations based on PZI-NPs and PAA (nPZI/P) and investigated whether the instillation of the newly developed nPZI/P can prevent postprandial hyperglycemia in a rabbit model involving the oral glucose tolerance test (OGTT).

## 2. Materials and Methods

### 2.1. Animals

The experiments used male Japanese adult rabbits (weight 2.64 ± 0.63 kg) and were performed according to the guidelines of the Association for Research in Vision and Ophthalmology and approved by Kindai University (KAPS-31-008, 1 April 2019). The rabbits were purchased from Shimizu Laboratory Supplies Co., Ltd. (Kyoto, Japan) and provided freely with water and CR-3 diet (Clea Japan Inc., Tokyo, Japan).

### 2.2. Chemicals

Humulin^®^N (CA-INS) was purchased from Eli Lilly Japan K.K. (Hyogo, Japan). Otsuka glucose injection was provided by Otsuka Pharmaceutical Co., Ltd. (Tokyo, Japan). MC was obtained from Shin-Etsu Chemical Co., Ltd. (Tokyo, Japan). Polyacrylic acid (PAA) was purchased from Wako Pure Chemical Industries, Ltd. (Osaka, Japan). Insulin ELISA Kits and Glucose Assay Kits were provided by Morinaga Institute of Biological Science Inc. (Kanagawa, Japan) and BioVision Inc. (Milpitas, CA, USA), respectively. Heat-inactivated fetal bovine serum (FBS), streptomycin, penicillin, and Dulbecco’s modified Eagle’s medium/Ham’s F12 (DMEM-F12) were purchased from GIBCO (Tokyo, Japan). Cell Count Reagent SF was obtained from Nacalai Tesque Inc. (Kyoto, Japan). All other chemicals used were of the highest purity commercially available.

### 2.3. Rabbit Model with Oral Glucose Tolerance Test (OGTT)

Rabbits were fasted for 15 h and then administered sucrose (1 g/kg). Blood samples were taken from a marginal ear vein at subsequent time points for the measurement of PG and INS concentrations. The INS ophthalmic formulations were instilled 3 min after the oral administration of sucrose.

### 2.4. Preparation of INS Ophthalmic Formulations

nPZI was prepared by the principle of rotation/revolution mixing (THINKY CORPORATION, Tokyo, Japan) of CA-INS using the Nano Pulverizer NP-100. A mixture of methylcellulose (MC), NaGC (bile salts, absorption enhancer of INS) and CA-INS was added into the tube together with zirconia beads (diameter: 0.1 mm), and milled by the Nano Pulverizer NP-100 (2000 rpm for 3 min × 3 times, 4 °C). The dispersions containing milled PZI were used as nPZI. In addition, nPZI/PL, nPZI/PM, and nPZI/PH were prepared by the addition of 0.001%, 0.01%, and 0.1% PAA, respectively (*w*/*v*%). Table 1 shows the composition of the INS ophthalmic formulations prepared in this study [20,21,22].

### 2.5. Characteristics in INS Ophthalmic Formulations

A laser diffraction particle size analyzer SALD-7100 (Shimadzu Corp., Kyoto, Japan) was used to measure the particle size with the refractive index set at 1.60-0.010i. The viscosity at 20 °C and the zeta potential of the INS ophthalmic formulations were measured by a SV-1A (A&D Company, Limited, Tokyo, Japan), and a micro-electrophoresis zeta potential analyzer model 502 (Nihon Rufuto Co., Ltd., Tokyo, Japan), respectively [21,22,23].

### 2.6. Dispersibility in INS Ophthalmic Formulations

The experiments were performed according to previous reports [21,22,23]. Briefly, 3 mL of ophthalmic formulations were added into tubes (total depth of the dispersions 4 cm) and incubated in the dark at 20 °C for 6 d. Then, 50 μL samples were collected from the upper 90% of the tube, mixed with rabbit blood serum, and the INS concentration was measured by an Insulin ELISA Kit according to the manufacturer’s instructions. In this study, the dispersibility was evaluated by measuring the INS concentration of the sample.

### 2.7. Corneal Toxicity of INS Ophthalmic Formulations

The immortalized human corneal epithelial cell line HCE-T [27] was cultured in DMEM-F12 containing 5% (*v*/*v*) FBS, 0.1 mg/mL streptomycin and 1000 IU/mL penicillin. 1 × 10^4^ HCE-T cells were seeded in 96-well microplates. At 24 h after seeding, the cells were treated with the INS ophthalmic formulations for 0–2 min. After that, Cell Count Reagent SF was added and incubated for 1 h to evaluate the cell damage. Then, the absorbance (Abs) at 490 nm was measured. Cell viability (%) was calculated as a ratio (Abs_treatment_/Abs_non-treatment_ × 100). The stimulation time was determined according to the in vivo retention time of the drugs in the cornea (about 2 min) [28].

### 2.8. Measurement of Plasma INS in Rabbits

Blood samples were obtained from a marginal ear vein of a rabbit, and the plasma was separated by centrifugation at 20,400× *g* for 20 min (4 °C). Plasma INS levels were determined using an Insulin ELISA Kit according to the manufacturer’s instructions, and ΔINS was estimated according to the following Equation (1):ΔINS = INS in rabbit with orally administered sucrose − INS in non-treated rabbit(1)

The area under the INS level time curve (AUC_ΔINS_) was calculated according to the trapezoidal rule from 0–240 min after the oral administration of sucrose.

### 2.9. Measurement of PG in Rabbits

Blood samples were obtained from a marginal ear vein of a rabbit, and the plasma was separated by centrifugation at 20,400 rpm for 20 min (4 °C). PG levels were determined using a Glucose Assay Kit according to the manufacturer’s instructions, and ΔGlu was estimated as follows:ΔPG = PG in rabbit with orally administered sucrose − PG in non-treated rabbit(2)

The area under the PG level time curve (AUC_PG_) and ΔPG (AUC_ΔPG_) were calculated according to the trapezoidal rule from 0–240 min.

### 2.10. Statistical Analysis

The data are expressed as the mean ± standard error (S.E.), and ANOVA followed by the Student’s *t*-test, and Dunnett’s multiple comparisons were used to analyze the differences between mean values. *p* < 0.05 was considered significant.

## 3. Results

### 3.1. Development of the INS Ophthalmic Formulations

Figure 1 shows the particle size frequencies of PZI in the INS ophthalmic formulations. Most of the PZI particles in CA-INS were in the micro-size range, however, the particle size of PZI was decreased by the mill treatment to the nano-size range (mean particle size in nPZI 197 ± 23 nm). In addition, the addition of PAA did not affect the particle size, with the mean particle size in PZI/PL, nPZI/PM and nPZI/PH of 203 ± 21 nm, 208 ± 17 nm, and 211 ± 18 nm, respectively. Figure 2 shows the viscosity, zeta potential, and dispersibility of the INS ophthalmic formulations. The viscosity of CA-INS was similar to that of water. On the other hand, the viscosity of nPZI was 2.17-fold that of CA-INS, and the viscosity was significantly increased by the addition of PAA (Figure 2A,B). The significant difference was not observed between the zeta potential of CA-INS and nPZI, however, the addition of PAA enhanced the zeta potential of PZI-NPs (Figure 2C,D). We also evaluated the dispersibility of PZI particles in the INS ophthalmic formulations (Figure 2E,F). The PZI particles precipitated quickly in CA-INS. In contrast, no aggregation of PZI particles was observed for 6 d in the nPZI and nPZI/P (nPZI/PL, nPZI/PM and nPZI/PH) formulations used in this study (Figure 2E,F). Figure 3 shows the corneal toxicity of the INS ophthalmic formulations. The viability of HCE-T cells treated with CA-INS was 98.0%, similar to that of cells treated with nPZI with or without PAA.

### 3.2. Changes in Plasma INS and PG Levels in Normal Rabbits Instilled with INS Ophthalmic Formulations

Next, we investigated whether the instillation of the INS ophthalmic formulations changed the plasma INS (Figure 4) and PG (Figure 5) in normal rabbits. There was only a slight increase in the plasma INS level in the rabbit instilled with CA-INS. In contrast, the instillation of nPZI resulted in a greater increase in the plasma INS level that peaked 10 min after instillation, and gradually decreased thereafter. The enhanced plasma INS levels were prolonged by the addition of PAA, with the AUC_ΔINS_ levels in rabbits instilled with nPZI/PL, nPZI/PM, and nPZI/PH 1.6-, 1.6-, and 2.5-fold that of nPZI, respectively (Figure 4). The PG levels decreased slightly, and then tended to increase in the rabbits instilled with nPZI and nPZI/P (nPZI/PL, nPZI/PM and nPZI/PH). On the other hand, there were no significant differences in the PG levels among the CA-INS, nPZI, and nPZI/P instilled rabbits (Figure 5).

### 3.3. Preventive Effect of the INS Ophthalmic Formulations on Blood Glucose Spike in the Rabbit OGTT Model

We next investigated whether the instillation of the INS ophthalmic formulations would attenuate the blood glucose spike in the rabbit OGTT model (Figure 6). The oral administration of sucrose enhanced the PG levels in the rabbits that peaked 40 min after the oral administration of sucrose and gradually decreased thereafter. The PG levels in the OGTT treated rabbits instilled with saline were 103.5 and 41.0 g/dl 40 min and 240 min after the oral administration of sucrose, respectively. Although the PG levels in the rabbit instilled with CA-IND were similar to saline-instilled rabbits, the AUC_ΔPG_ for rabbits instilled with nPZI were significantly lower than rabbits instilled with saline or CA-IND. In addition, the combination of PZI-NPs and PAA (nPZI/P) attenuated the enhancement of PG levels further in comparison with nPZI.

## 4. Discussion

Postprandial hyperglycemia, the so-called blood glucose spike, is associated with enhanced risks of DM and its complications [4,5,6,7]. Reducing postprandial PG is the main target in the treatment and prevention of DM and DM complications [11]. In this study, we designed ophthalmic formulations based on PZI-NPs and PAA (nPZI/P) and found that pre-instillation of nPZI/P could improve postprandial hyperglycemia by INS delivered through the ocular route in the rabbit model.

It would be useful if ophthalmic formulations could enhance the therapeutic efficacy of INS delivered by an ocular DDS [26]. However, the ocular instillation of traditional INS results in low BA [13,17] and short duration [18] due to extensive drug loss at the ocular surface. Therefore, it is necessary to improve the absorption and increase the INS contact time at the absorption site. There have been several studies in which the use of a gel system has been reported to improve ocular BA and prolong the precorneal resident time of INS. Yamamoto et al. [26] reported that BA via the ocular route of an INS solution containing 1% NaGC, which is an absorption enhancer, was approximately 1%. However, the addition of PAA enhanced the BA to greater than 5% for an INS solution comprising 1% NaGC and 0.1% PAA. In particular, PAA is more effective in increasing the ocular BA of INS in comparison with other gel bases such as Na-glycocholate, hyaluronic acid, or polyvinylalcohol [26]. In addition, we previously reported that the corneal and conjunctival accumulation of an ocular drug can be enhanced by adjusting the particle size of the drug to a range of 140–150 nm [29]. Although INS is soluble, PZI, which is a type of man-made INS, is a crystalline suspension that has been accepted to increase the medical safety of INS [30]. Based on these findings, we attempted to prepare ophthalmic formulations based on PZI-NPs and PAA (nPZI/P). After bead mill treatment, the particle size of the PZI was in the range of 80–550 nm (Figure 1), with no aggregation observed for 6 d (Figure 2E,F). It is known that viscosity and zeta potential are related to particle aggregation. While the viscosity of CA-INS is similar to that of water, the viscosity of nPZI was found to be 2.17-fold higher, and this was significantly increased by the addition of PAA (Figure 2A,B). On the other hand, the zeta potentials of nPZI were similar to that of CA-INS (Figure 2C,D). These results suggest that the decreased particle size and enhanced viscosity prevent the aggregation of INS-NPs, resulting in an enhancement of dispersibility in comparison with CA-INS.

Next, we investigated changes in the plasma INS and PG levels following the instillation of ophthalmic INS formulations in normal rabbits (Figure 4 and Figure 5). The plasma INS levels in normal rabbits instilled with nPZI were significantly higher than in rabbits instilled with CA-INS (Figure 4A,B) and the AUC*_Δ_*_INS_ in the nPZI was higher in comparison with the CA-INS with 0.1% PAA (191 ± 23.5 ng/min·min, *n* = 4). However, while PG levels tended to decrease by the instillation of nPZI, the difference in the PG levels between normal rabbits instilled with CA-INS and nPZI was not significant (Figure 5A,B). We previously reported that particles approximately 150 nm in size are taken up into the eye by multiple energy-dependent endocytosis, resulting in increased ophthalmic absorption [21]. Therefore, our results support the previous study [21]. On the other hand, the enhanced plasma INS levels provided by nPZI were not enough to decrease PG, since the instillation of nPZI did not cause a rapid decrease PG under normal conditions. Therefore, a single instillation of nPZI may not cause hypoglycemia and be safe to use under normal conditions. Next, we investigated whether nPZI could prevent postprandial hyperglycemia in the rabbit model using OGTT (Figure 6). Although the AUC_ΔPG_ was similar in rabbits instilled with saline and CA-INS, the AUC_ΔPG_ for rabbits instilled with nPZI was 73.2% that in rabbits instilled with saline. Thus, nPZI may be effective in preventing postprandial hyperglycemia. Further work is needed to enhance the BA of INS delivered via the ocular route for practical use. Therefore, we also demonstrated the preventive effect of nPZI/P on postprandial hyperglycemia in the same rabbit OGTT model (Figure 6). The rise in PG in the rabbit OGTT model was further attenuated by the addition of PAA to the formulation. These results support the data for plasma INS levels in rabbits instilled with nPZI/P (Figure 4). The PG levels in normal rabbits tended to decrease in the 0–30 min time frame following the instillation of nPZI/P, although the difference between PG levels in rabbits instilled with nPZI/P and saline was not significant (Figure 5C,D). Such normalization of PG may be due to the regulation of INS secretion since PG levels are closely regulated in healthy individuals. In addition, it is known that the enhancement in the PG level that occurs postprandially seldom lasts beyond 3 h in normal patients due to the induction of INS (positive feedback). Taken together, we hypothesize that nPZI/P has no effect on PG levels in normal rabbits but can suppress the rapid changes in PG levels that occur in postprandial hyperglycemia.

A further study to evaluate the biodstribution of nPZI/P after the instillation is needed. In addition, it is important to clarify the mechanism of INS absorption and the topical effect of nPZI/P in the case of ophthalmic disease, such as corneal epithelial erosion in DM. In future work, we plan to investigate the therapeutic effect of nPZI/P in a DM model, such as Otsuka Long-Evans Tokushima Fatty rat, streptozotocin-induced diabetic rats, or Goto-Kakizaki rats.

## 5. Conclusions

We designed PZI-NPs by the bead mill method and prepared ophthalmic formulations based on PZI-NPs and PAA (nPZI/P). Moreover, we showed that the nPZI/P formulation had no effect on PG levels in normal rabbits, while the rapid enhancement in PG levels during postprandial hyperglycemia was attenuated by a single instillation before the OGTT. Our studies are the first in which postprandial hyperglycemia (blood glucose spike) was prevented by an ophthalmic formulation based on solid INS-NPs and PAA. These findings suggest a possibly better way to manage the blood glucose spike, which is an important target in type 2 DM therapy.

## Figures and Tables

**Figure 1 pharmaceutics-13-00375-f001:**
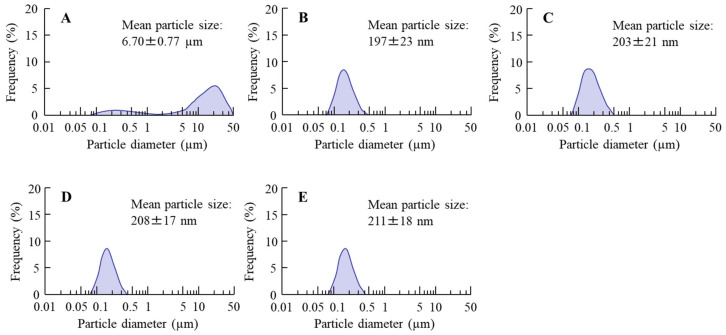
Particle size frequencies of PZI in CA-INS (**A**), nPZI (**B**), nPZI/PL (**C**), nPZI/PM (**D**), and nPZI/PH (**E**). The compositions of the INS ophthalmic formulations are shown in Table 1. The particle size of PZI was decreased by the bead mill, and the particle size frequencies in nPZI and nPZI/P were approximately 80–550 nm.

**Figure 2 pharmaceutics-13-00375-f002:**
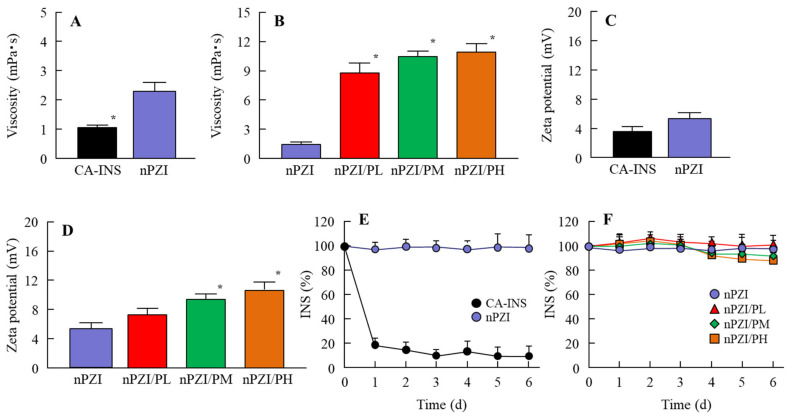
Viscosity (**A**,**B**), zeta potential (**C**,**D**) and dispersibility (**E**,**F**) of PZI in CA-INS, nPZI, nPZI/PL, nPZI/PM, and nPZI/PH. The compositions of the INS ophthalmic formulations are shown in Table 1. The dispersibilities of PZI particles in the INS ophthalmic formulations were measured 6 d after bead mill treatment. *n* = 8. * *p* < 0.05 vs. nPZI for each category. The viscosity of nPZI was higher than that of CA-INS, and the viscosity was increased by the addition of PAA. No aggregation of the PZI particles in nPZI and nPZI/P was observed for 6 d.

**Figure 3 pharmaceutics-13-00375-f003:**
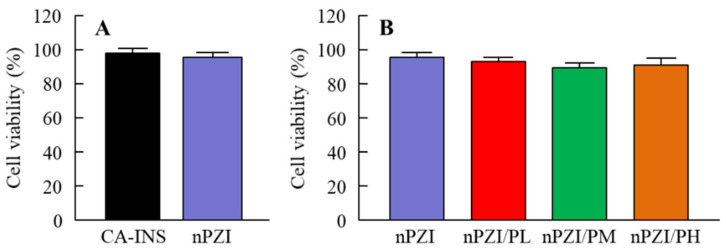
Corneal toxicity (**A**,**B**) in HCE-T cells treated with CA-INS, nPZI, nPZI/PL, nPZI/PM, or nPZI/PH. The compositions of the INS ophthalmic formulations are shown in Table 1. *n* = 8. Corneal toxicity was not observed for the nPZI and nPZI/P used in this study.

**Figure 4 pharmaceutics-13-00375-f004:**
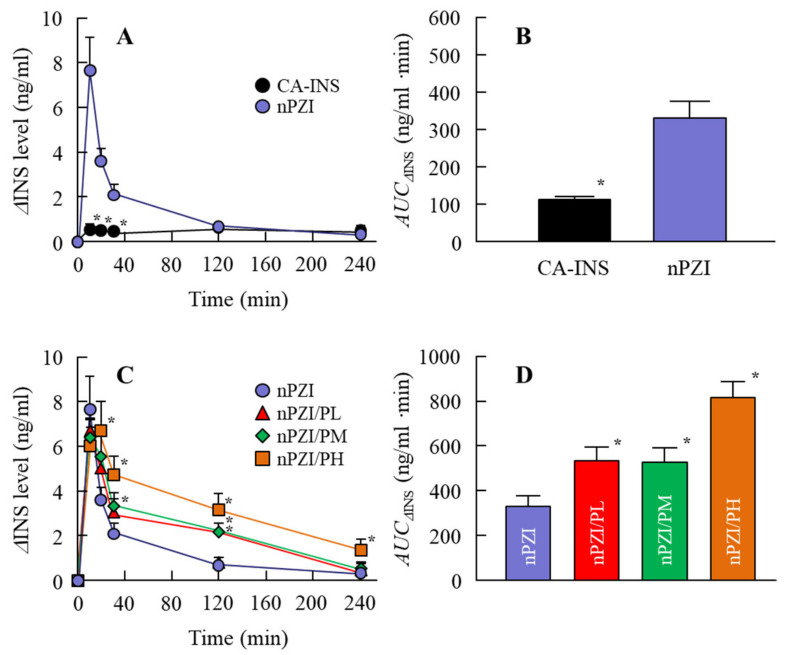
Changes in plasma INS (**A**,**C**) and AUC_ΔINS_ (**B**,**D**) levels in normal rabbits instilled with CA-INS, nPZI, nPZI/PL, nPZI/PM, or nPZI/PH. The compositions of the INS ophthalmic formulations are shown in Table 1. *n* = 6–8. * *p* < 0.05 vs. nPZI for each category. The ΔINS levels in rabbits instilled with nPZI were significantly higher than in rabbits instilled with CA-INS, and the ΔINS levels were enhanced by the addition of PAA in nPZI/P.

**Figure 5 pharmaceutics-13-00375-f005:**
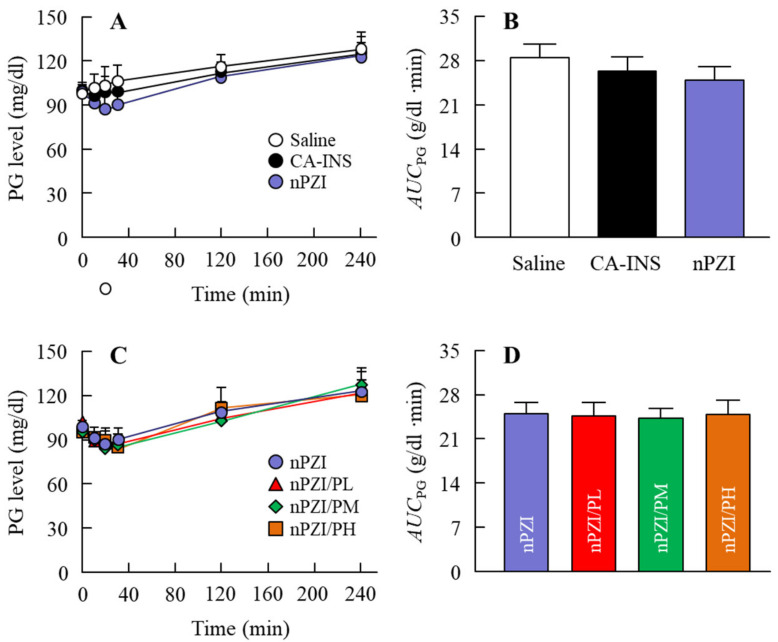
Changes in PG (**A**,**C**) and AUC_PG_ (**B**,**D**) levels in normal rabbits instilled with CA-INS, nPZI, nPZI/PL, nPZI/PM, or nPZI/PH. The compositions of the INS ophthalmic formulations are shown in Table 1. *n* = 6–8. No significant differences in PG levels were observed among normal rabbits instilled with CA-INS, nPZI, or nPZI/P.

**Figure 6 pharmaceutics-13-00375-f006:**
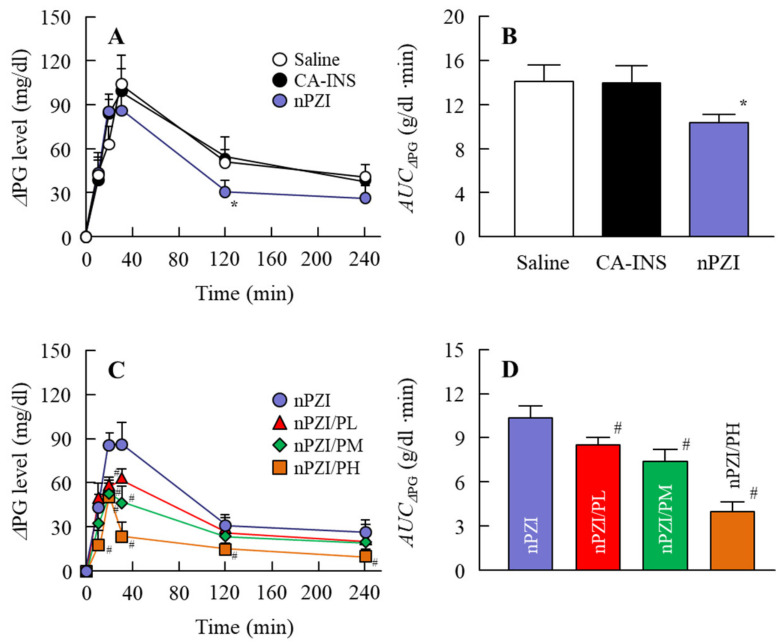
Changes in ΔPG (**A**,**C**) and AUC_ΔPG_ (**B**,**D**) levels in the rabbit OGTT (oral glucose tolerance test) model in rabbits instilled with CA-INS, nPZI, nPZI/PL, nPZI/PM, or nPZI/PH. The compositions of the INS ophthalmic formulations are shown in Table 1. *n* = 6–8. * *p* < 0.05 vs. Saline for each category. ^#^
*p* < 0.05 vs. nPZI for each category. The postprandial hyperglycemia in rabbits instilled with nPZI was significantly attenuated in comparison with rabbits instilled with CA-INS, and the preventive effect was enhanced by the PAA content in the nPZI/P.

**Table 1 pharmaceutics-13-00375-t001:** Compositions of INS ophthalmic formulations.

Formulation	PZI	NaGC	MC	PAA	Treatment
PZI-NPs	0.3%	1%	4%	—	Bead mill
PZI/PL	0.3%	1%	4%	0.001%	Bead mill
PZI/PM	0.3%	1%	4%	0.01%	Bead mill
PZI/PH	0.3%	1%	4%	0.1%	Bead mill

The data are expressed as the *w*/*v*%.

## Data Availability

Not applicable.

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
