# Peer review of "Prevention of Postprandial Hyperglycemia by Ophthalmic Nanoparticles Based on Protamine Zinc Insulin in the Rabbit"

_pharmaceutics, 2021, doi:10.3390/pharmaceutics13030375_

Round 1
Reviewer 1 Report
In general, submitted manuscript is very well written and would be of great interest to the readers of Pharmaceutics. I have few points below which authors may want to look into.
- Introduction:
- Line 68 - 'significant increase in...'
- Please add the rationale behind using bead mill specifically. What is a bead mill (schematic, if required?)? Why did you choose this and any previous references to justify the suitability?
- Material and method:
- Supplier and other information of NaGC
- Define MC
- Equations 1 and 2 – ‘rabbit with…’
- Results and discussion:
- I do not doubt the effect of smaller particle size in the improvement of BA, but I wonder if addition of PAA to CA-INS alone could have significant impact on the BA. Although PS of CA-INS is quite high, but I think authors should add a rationale behind not doing this. It could simply be a statement or two based around the issue with large PS or a reference if available.
Author Response
We carefully revised our manuscript according to the suggestions of the reviewer 1, and details are as follows.
< Q and A for Reviewer 1>
Q1. Introduction: Line 68 - 'significant increase in...' Please add the rationale behind using bead mill specifically. What is a bead mill (schematic, if required?)? Why did you choose this and any previous references to justify the suitability?
A1. Thank you very much for pointing this out. Previous research over several decades focused on two major approaches to the design of nanoparticles: bottom-up synthesis and a top-down approach. The bead mill method is one of the top-down approach, and is applied to a wide range of applications, such as grinding relatively large particles and dispersing nanoparticles. On the other hand, it is necessary to select appropriate additives for nanoparticulation by using the bead mill method. In this study, we have cited our previous reports (Ref. 20-25) as evidence of these basic experiments. In order to respond to the reviewer’s comment, we add the explanation of the bead mill in the Introduction (line 60-61).
Q2. Material and method: Supplier and other information of NaGC,
Define MC
Equations 1 and 2 – ‘rabbit with…’
A2. The reviewer’s comment is correct. The NaGC (bile salts) was used to enhance the absorption of insulin, and MC is abbreviation of methylcellulose. In order to respond to the reviewer’s comment, we added the contents, and revised to “INS in rabbit with orally administered sucrose” in the equation (line 106, 147, 158, 259).
Q3. Results and discussion: I do not doubt the effect of smaller particle size in the improvement of BA, but I wonder if addition of PAA to CA-INS alone could have significant impact on the BA. Although PS of CA-INS is quite high, but I think authors should add a rationale behind not doing this. It could simply be a statement or two based around the issue with large PS or a reference if available.
A3. The reviewer’s comments are very important. The AUCDINS in the CA-INS with 0.1% PAA was 191±23.5 ng/min∙min (n=4). In order to respond to the reviewer’s comment, we added the data in the result (line 280-281).
Thank you for great comments.

Reviewer 2 Report
The authors have shown how the effect of the NPZI variations affect the INS and PG in normal and "diseased" rabbits. The results are very interesting the the authors have done a good job showing the effectiveness of the treatments. I would suggest several changes.
in section 2.4 where you describe the formulations. are the % in w:volume.... it is difficult as no volume was stated and therefore the methodology is incomplete to understand the procedure.
2.7 while I can understand that the BA in the eye is 2 minutes in vivo, it seems obvious that testing in vitro toxicity at 2 minutes will show no results........ is there any other precedence for testing this rapidly? in vivo, is it assumed that after 2 minutes the compounds have alreaedy entered into the blood stream and therefore are no longer available for toxicity at the site of injection? please explain
In addition to the previous point. is there any biodstribution data as to what happens to these types of nps in vivo? obviously they do not remain at the injectino site (2 min) where do they go and what are the potential implications?
Author Response
We carefully revised our manuscript according to the suggestions of the reviewer 2, and details are as follows.
< Q and A for Reviewer 2>
Q1. in section 2.4 where you describe the formulations. are the % in w:volume.... it is difficult as no volume was stated and therefore the methodology is incomplete to understand the procedure.
A1. The reviewer’s comment is correct. The data are expressed as the w/v%. In order to respond to the reviewer’s comment, we added the information in the Materials and Methods (line 110, 114).
Q2. 2.7. while I can understand that the BA in the eye is 2 minutes in vivo, it seems obvious that testing in vitro toxicity at 2 minutes will show no results........ is there any other precedence for testing this rapidly? in vivo, is it assumed that after 2 minutes the compounds have already entered into the blood stream and therefore are no longer available for toxicity at the site of injection? please explain.
A2. The reviewer’s comments are very important. In general, the eye drops was diluted by the lacrimal fluid (approximately 10-fold dilution) immediately after the instillation, and drugs is flowed through the nasolacrimal duct. On the other hand, we used the non-diluted eye drops in this in vitro study, and treated for 2 min. Therefore, it was suggested that the stimulation in the in vivo study was higher than that in the in vivo condition. Taken together, we determined the protocol following; the cells were treated with the INS ophthalmic formulations for 2 min. After that, Cell Count Reagent SF was added, and incubated for 1 h to evaluate the cell damage. Then, the absorbance (Abs) at 490 nm was measured. In order to respond to the reviewer’s comment, we added the content in the Materials and Methods (line 135-137).
Q3. In addition to the previous point. is there any biodstribution data as to what happens to these types of nps in vivo? obviously they do not remain at the injectino site (2 min) where do they go and what are the potential implications?
A3. Thank you very much for pointing this out. Our previous study using cilostazol nanoparticles (Nagai et al., Pharm Anal Acta, 2015, 6:4) showed that the nanoparticles was taken into the cornea and conjunctiva, and dissolved in the tissue. After that, the dissolved-drug were delivered to blood, lens, sclera, choroid and retina. Thus, the drugs was diluted by the lacrimal fluid immediately after instillation, and after that, the drug on the ocular surface begins to shift to other tissues. On the other hand, we don’t have the biodstribution data of the nPZI/P. In order to respond to the reviewer’s comment, we added the importance as further study in Discussion (line 308-309).
Thank you for great comments.
